# Large-Scale Compatible Roll-to-Roll Coating of Paper Electrodes and Their Compatibility as Lithium-Ion Battery Anodes

**DOI:** 10.3390/nano15020113

**Published:** 2025-01-14

**Authors:** Nicklas Blomquist, Manisha Phadatare, Rohan Patil, Renyun Zhang, Noah Leuschen, Magnus Hummelgård

**Affiliations:** Department of Engineering, Mathematics and Science Education, Mid Sweden University, SE-851 70 Sundsvall, Sweden; manisha.phadatare@miun.se (M.P.); rohan.patil@miun.se (R.P.); renyun.zhang@miun.se (R.Z.); magnus.hummelgard@miun.se (M.H.)

**Keywords:** paper electrodes, sustainable, recyclable, resource efficient, graphene, nanographite, nanoplatelets, cellulose binder, energy storage, lithium ion

## Abstract

A recyclability perspective is essential in the sustainable development of energy storage devices, such as lithium-ion batteries (LIBs), but the development of LIBs prioritizes battery capacity and energy density over recyclability, and hence, the recycling methods are complex and the recycling rate is low compared to other technologies. To improve this situation, the underlying battery design must be changed and the material choices need to be made with a sustainable mindset. A suitable and effective approach is to utilize bio-materials, such as paper and electrode composites made from graphite and cellulose, and adopt already existing recycling methods connected to the paper industry. To address this, we have developed a concept for fabricating fully disposable and resource-efficient paper-based electrodes with a large-scale roll-to-roll coating operation in which the conductive material is a nanographite and microcrystalline cellulose mixture coated on a paper separator. The overall best result was achieved with coated roll 08 with a coat weight of 12.83(22) g/m^2^ and after calendering, the highest density of 1.117(97) g/cm^3^, as well as the highest electrical conductivity with a resistivity of 0.1293(17) mΩ·m. We also verified the use of this concept as an anode in LIB half-cell coin cells, showing a specific capacity of 147 mAh/g, i.e., 40% of graphite’s theoretical performance, and a good long-term stability of battery capacity over extended cycling. This concept highlights the potential of using paper as a separator and strengthens the outlook of a new design concept wherein paper can both act as a separator and a substrate for coating the anode material.

## 1. Introduction

The rapid growth of electric vehicles (EVs) and growing concern over global warming, reducing the carbon emissions associated with the entire life cycle of lithium-ion batteries (LIBs) have become a central focus for various entities including research institutions across the globe [1]. Developing LIBs with a recyclability perspective is essential for reducing carbon dioxide emissions. However, today’s lithium-ion batteries prioritize battery capacity and energy density over recyclability, and hence, lithium batteries are currently being recycled at rates less than 50%, as compared with lead–acid batteries that have a recycling rate of 99.5% [2]. When it comes to recycling, the emphasis should be on developing recycling methods that allow the entire battery to be recycled. However, most of efforts are being made to improve recycling methods associated with cathode materials, while other critical components such as electrolytes that are often toxic to the environment do not receive any major attention, unfortunately [3]. Step utilization and recovery separation reproduction are the two approaches used to increase sustainability; in short, the former makes more efficient use of existing batteries, and the latter is a cluster of different types of recycling methods, i.e., pyro-, hydro-, and biometallurgy, and direct regeneration [3]. Neither of these methods is ideal and all have some major drawbacks; pyrometallurgical processes are simpler but suffer from not being able to recover the lithium metal and releasing hazardous gases [4]. Hydrometallurgical treatment, which is the most popular one today with more than half of all processes sorted under this category, also allows for lithium recovery but is more complex than the pyrometallurgical process [4]. To achieve good efficiency in recovering the metals, it has been proposed that hydrometallurgy should be combined with biometallurgy, followed by a second step of bioleaching with bacteria or funguses to recover the metals and boost the efficiency of the recovery process [3]. The drawback of the bioleaching process is that as the concentrations of minerals in the waste material go up, the bacteria’s efficiency decreases due to the toxic effects [4]. The direct regeneration method implies that materials are not first recycled but, in some way, directly reused, i.e., the cells are repacked with old electrode materials, and although it is a simple and resource-effective approach, the drawbacks are contamination and performance issues of the regenerated cells [5]. The issues being mentioned are caused by design flaws in batteries where the focus has been on developing cells with high performance and high economical return. However, from a sustainable perspective, the development lacks the two remaining factors of high environmental benefit and high safety [6]. A sustainable-perspective overview of the components of a regular lithium-ion battery presents several issues. The current collectors, made from aluminum and copper foils, are produced via high-energy-consuming processes, and the cathode materials are high in cost, non-renewable, and contain toxic elements. The binders used are fluorinated versions, and electrolytes such as LiPF6 are toxic and unsafe in decomposition scenarios. The anode materials are graphite versions from non-renewable sources that contribute to either pollution or high energy consumption in their production. Lastly, the separator is made from plastic, is not thermally stable, i.e., a safety issue, and is produced via mineral oils [7].

As the quote says, if the shoe fits, so it is the underlying battery design that must be changed to more efficiently improve the situation. A new shoe, or rather a sustainable battery design, is to a high degree dominated by material choices. These materials must be able to fulfill these high environmental benefit and high safety criteria and do so at their core level and all the way throughout the value chain. A new and more effective approach is to utilize bio-materials in battery design more frequently, and there are many publications on electrodes where the fossil or synthetic graphite is replaced by biocarbon products. These are much more resource-efficient and can be generated from biomass or biowaste materials; lignin and cellulose are two sources for biocarbon generation [8]. These biomass materials also contain other biomolecules that in the long run can function as the active components in the redox reactions of the battery instead of relying on metals, making a completely metal-free battery [9]. The metallic current collector itself can also be replaced with carbon alternatives, eliminating the energy consumption of metal foil production; one way of achieving this is by the direct carbonization of a cellulose paper substrate in a furnace, or one can use laser processing tools to carbonize the top layer of a paper directly [10,11]. Using biobased recycling methods such as fungi or bacteria for recycling is an overall good and sustainable approach, as long as the environment the organisms operate in is suitable; this leads to less toxicity as well as good organism penetration and accessibility into the battery during decomposition due to the reduced usage of metals, foils, and plastics, which is needed to improve their overall slow performance [12]. This concept can be extended further by allowing the organisms to construct the new electrode material as well; this is referred to as biofabrication [13]. Hence, many of the components in a battery can be built around different types of paper substrates, and this gives us the concept of paper-based batteries, or paper-based energy storage devices in general, and these have potential to provide many solutions to the sustainability problems of existing battery technology. As described above, a cellulose-based material is renewable, resource-efficient, and non-toxic, has a fully developed recycling platform in its value chain, and lastly, it also has an existing industry capable of producing the large volumes of materials needed for a green societal transition via battery technology implementation [14]. Paper can serve as a separator instead of plastic, and it can be converted into electrodes by different coating techniques. It can achieve current collector functionality through carbonization methods, and since the material is flexible, new types of packaging and encapsulating techniques can also be introduced, such as folding [15]. LIB polypropylene (PP) or polyethene (PE) separators are usually less than 25 μm thick sheets with 40% porosity and sub-micrometer pore sizes. Using paper as a separator has been shown to be advantageous even in terms of performance, as these have proven MacMullin numbers (ratio of ion conductivity between a wetted separator and the ion conductivity of the free electrolyte) of 3–6 [16], compared to the typical 20 for a PE separator [17], where a lower ratio corresponds to better ion conductivity. To make a paper-based electrode, the paper itself can be converted into an electrode, and this has already been demonstrated in pilot-scale operations in which paper was fabricated containing active carbon and PEEDOT:PSS ingredients [18,19]. However, most often, regular coating techniques or methods are used, such as printing, casting/filtration, or thermal evaporation to deposit an electrically conductive material on top of the paper to serve as the electrode. Commonly, different types of nanomaterial composites with carbon nanotubes, graphene, or similar are used, and in this scenario, paper obtains additional functionality to serve as a separator as well [20]. Adding conductive materials such as carbon nanotubes on top of the paper via Meyer rod coating has already demonstrated excellent results, such as achieving a surface resistivity of 1 ohm per square, and realized devices such as supercapacitors with specific capacitance of 200 F/g [21]. Another reported method is the spray deposition of graphite and microfibrillated cellulose onto bleached softwood pulp prior to pressing and drying in a pilot paper machine. This method achieves an LIB anode capacity of 95 mAh/g at 1 C with an electrode thickness of 27.5 μm, specific weight of 11.6 g/m^2^, and electrical resistivity of about 500 Ω/sq (about 14 Ω·m) [22]. Besides paper electrodes being used for supercapacitor purposes, lithium-ion paper batteries have already been demonstrated with cellulose binders in a LiFePO_4_ cathode and graphite anode [14,23]. Even more complex electronic devices can be realized with paper; for example, printing allows us to print a battery on a local area of the paper, which can be integrated with other surrounding components [24]. In addition to energy storage applications, our own studies have shown that paper with conductive coatings can also be used for energy harvesting in triboelectric nanogenerators with power densities exceeding 14 kWm^−2^ [25].

Here, we report the results of fabricating a fully disposable and resource-efficient paper-based electrode with a large-scale roll-to-roll coating operation in a paper pilot facility at speeds of up to 25 m/min in which the conductive material is a graphene/graphite mixture (nanographite) with microcrystalline cellulose (MCC) as the binder. The nanographite was fabricated in-house by our earlier-developed water-based large-scale compatible exfoliation technique [26,27]. The produced electrode material was then verified in a typical lithium-ion half-cell coin cell setup in which the electrode paper acted as both an anode and separator, together with standard lithium foil as the cathode, and default LP40 was used as the electrolyte.

## 2. Materials and Methods

### 2.1. Electrode Material Preparation

In this study, two different coating colors (electrode material suspensions) were used, called Slurry A and Slurry B. Both coating colors had the same amount of MCC addition as the binder but different nanographite sources. For Slurry A, a 1000 L nanographite suspension (GS14, SN:1013-1052) with 40 gL^−1^ solids content was purchased from 2Dfab in Sundsvall, Sweden. This nanographite was based on graphite raw material from Imerys and a Pluronic dispersion agent. The nanographite suspension was de-watered to 95 gL^−1^ solids content by removing the clear water phase above the sedimented nanographite. For Slurry B, a 400 L nanographite suspension was fabricated in-house by our own water-based large-scale compatible exfoliation technique, described earlier by Blomquist et al. [27]. The shear zone used was a 2 mm helical coil tube, referred to as S2 in the original method. The solids content of graphite in the suspension was increased to 75 gL^−1^ and the flow rate was held constant at 5 Lmin^−1^ for 10 full passes through the shear zone. The graphite used was thermally expanded natural crystalline graphite (EXG 9840) from Graphit Kropfmühl in Passau, Germany with an addition of 2 wt% polyacrylic acid (Sigma-Aldrich, St. Louis, MO, USA), in relation to the graphite mass, as the dispersant. For additional details, see Appendix A. The MCC used as the binder was Refined MCC (White MCC) from Fibenol in Tallinn, Estonia. The MCC was delivered as a paste with 18.5% solids content and was used as is without further modifications. In both Slurry A and Slurry B, 10 wt% MCC was added to the nanographite suspensions, recalculated as dry weight in relation to the dry content of nanographite. The nanographite suspensions with MCC were mixed using a Cowles mixer to achieve uniform coating colors. The volume and solids contents of the final coating colors were 440 L at 10 wt% for Slurry A and 412 L at 8 wt% for Slurry B.

### 2.2. Electrode Coating

Two different substrate papers were used in this study: Advantage Kraft Plus 70 g (Plus), with a measured grammage of 68.54(22) g/m^2^ and a thickness of 96.60(50) μm, and Advantage Boost HP 80 g (Boost), with a measured grammage of 78.87(19) g/m^2^ and a thickness of 124.83(45) μm, both Kraft papers and both from Mondi Dynäs in Kramfors, Sweden. The substrates were selected based on their high dry and wet tensile strength and porosity. The main difference in properties, except the grammage, between the substrates is air permeability; for Advantage Boost HP, no hydrophobing agent is added in production, leading to higher permeability. The substrate papers were delivered as 500 kg rolls with a diameter of 1200 mm and a width of 520 mm. The rolls were used as is, and the coating was applied on the outer slightly rougher paper side of the rolls. The coating was performed at the UMV Coating Systems Pilot plant in Säffle, Sweden, using the UMV Liquid Application System (LAS). In short, this coating applicator transferred the coating color with an applicator roll from a pan to a metering nip. The metered film of coating color was then applied onto the substrate paper by a hydrophilic transfer roll. The variable coating parameters were machine speed, transfer roll speed, and metering nip, as well as individual control of a series of IR dryers and hot air dryers. See Appendix A for images of the coating operation.

### 2.3. Sample Preparation and Characterization

Samples from the coated rolls were cut both across the width of the roll and lengthwise according to Figure 1.

For each roll, the following samples were taken: Three sample strips with a width of 2–3 cm were cut across the full width of the roll; the two first strips were cut 10 cm apart from each other, and the third strip was cut 1 m away from the first strip; these are referred to as w1, w2, and w3. Another set of three sample strips, 1 m long and also about 2–3 cm wide, were cut in the lengthwise direction of the roll from three different sections of the roll width. The first of these sample strips was positioned 5.5 cm from the edge (edge-facing operator), the second was positioned 29 cm from the edge (the middle section), and the third sample strip was positioned 49 cm from the edge. These sample strips are referred to as l55, l29, and l49. Prefixes to notation correspond to the roll or sheet sample and roll number; for example, roll-03B-l55 refers to roll sample 03B, a lengthwise section 5.5 cm from the edge.

#### 2.3.1. Electrical and Physical Measurements

For each sample strip, the electrical resistance was measured between the edge and multiple points along the length of the strip. The measurements were made in stepwise length increases of 1 cm with a Tillquist TQ711 multimeter (Kista, Sweden), and the resistance dependency as a function of length was determined with linear regression. The electrical bulk resistivity (ρ) of the coating was then calculated from the slope factor of the regression, as given by Equation (Equation 1): (1)$$R=\frac{\rho }{WH}\cdot L,$$
in which *R* is the measured electrical resistance, *W* is the width of the strip, *H* is the coating thickness, and *L* is the length of the strip. To measure the coating thickness and calculate coat weight, sets of 100 coins with a diameter of 16 mm were punched out from the sample strips including one strip of uncoated substrate paper as a reference. The thickness was measured with a Mitutoyo N0.2046 (0.01–10 mm, Kawasaki, Japan) analog micrometer gauge fitted with a spring-loaded elephant foot, and the mass measurements were performed with a Mettler Toledo XS204 Analytical Balance (Columbus, OH, USA) with a readability of 0.1 mg and a repeatability of 0.07 mg. Further details are shown in Appendix A. The electron microscopy imaging was performed with a field emission scanning electron microscope (TESCAN MAIA3-2016, Brno, Czech Republic) at 1 kV–3 kV with an SE detector. The top-view images are oriented so the lengthwise paper direction is upwards in the image. Simple light transmission tests were performed by placing the sample strips on a light table and studying the variance of transmitted light across the surface. Tape tests were used to study nanographite flake adhesion in the coating by placing and pulling Scotch tape strips on the top of the coating for each roll.

#### 2.3.2. Lithium-Ion Anode Application

LIB half-cells were assembled as CR2025 coin cells under an argon atmosphere in a glove box. Discs of the coated paper with a 16 mm diameter were used as the anode and separator, wherein the paper side acted as a separator and the coating as an anode. Another 16 mm disc of copper foil was added as a current collector on the nanographite (anode) side, to allow comparison with traditional reference cells without influence from different electrode–contact interface materials. Three cells B3-1, B3-2, and B4-2 were fabricated with the Advantage Plus paper and from roll 03B. Two cells, B5-2, and B6-2, were made with Advantage Boost paper from roll 11. A 0.2 mm thick and 13 mm diameter disc of lithium metal foil from Goodfellow in England was used as a reference and counter-electrode (cathode). LP40 was used as an electrolyte (1M LiPF6 in a mixture of ethylene carbonate and diethyl carbonate in a 1:1 weight ratio). Cyclic voltammetry (CV) tests of the cells were performed between 0.01 and 1.5 V at a scan rate of 0.1 mVs^−1^. Galvanostatic charge–discharge (GCD) tests of the cells were performed at a current density of 100 mAg^−1^ in a voltage range between 1 mV and 1.0 V. The specific capacities and current densities were calculated based on the weight of the active materials of the electrode. All of the electrochemical measurements were conducted at room temperature. To study the ionic conductivity effects of the paper separator, an additional set of four coin cells, 4:1, 4:2, 4:3, and 4:4, were fabricated in a similar way but with additional uncoated papers inserted as extra separator materials, i.e., 4:1 contained one extra paper, 4:2 two and so on. The thickness of the uncoated paper sheets was 96.60(50) μm. These four cells were all made with Advantage Plus and roll 03B, and were then characterized for electric series resistance (ESR) performance with an Ametek Parstat potentiostat at both 1 mV DC (discharged state) and 1.5 V DC (charged state) during a traditional frequency sweep from 1 MHz to 100 mHz at a 10 mV AC amplitude. The resistivity of the paper separator was then calculated from the linear regression of ESR values from low-end frequency vs. total separator thickness. For comparison with the paper-based cells, a traditional reference cell was fabricated, cell R1, in which Slurry A was cast onto copper foil instead of paper at lab-scale. As a separator, a commercial Cellgard 2325 plastic membrane was used, and besides this, the rest of the assembly sequence as well as the electrolyte and lithium was the same as for the other paper-based cells.

## 3. Results

Table 1 shows a summary of the paper electrode rolls and their respective slurry, paper substrate, coat speed, transfer roll (TR) speed, metering nip, final coat weight, coating thickness, and calculated coating density.

The maximum coating thickness obtained during the stable coating operation with Slurry A was from roll 03B with a coat weight of 4.39(31) g/m^2^ and a coating thickness of 16.4(1.0) μm. With higher coat weight, the coating operation became unstable, resulting in frequent paper failure, starting with wrinkling followed by paper breakage. Repeated coating on the same substrate paper (Plus) with Slurry B resulted in roll 06 and roll 07, differentiated by adjustment of the transfer roll speed. Roll 07 obtained a coating thickness of 46.8(2.1) μm at a coat weight of 11.42(31) g/m^2^ during the stable coating operation. With identical settings as in roll 07, coating Slurry B on the Boost paper substrate resulted in roll 08 with a coat weight of 12.83(22) g/m^2^ and a corresponding coating thickness of 25.9(1.1) μm. Roll 08 indicates that increased coat weight generates a higher density coating, suggesting a calendering effect of the increasing paper web tension when the coating faces a steel roll prior to winding. Further attempts were made to increase coat weight with maintained density by lowering the machine speed and transfer roll speed and thus allowing for decreased web tension; this resulted in roll 10 to roll 12. Roll 10 had insufficient drying and was still wet during winding. The maximum coat weight achieved was 17.65(29) g/m^2^ with Slurry B and Boost substrate paper for roll 11; the corresponding coating thickness was 47.7(1.4) μm. Roll 12 was made by repeating the setting of roll 06 but on the Boost substrate paper, generating a coat weight of 10.79(24) g/m^2^ and a coating thickness of 56.1(2.5) μm. Rolls missing from the chronological numbering were either uncoated rolls used for paper samples or rolls made only to adjust the coating parameters to achieve stable coating operation. To obtain an indication of the influence of density on the physical and electrical properties, sheets cut from rolls with Slurry B were calendered without heat and included in the following results.

### 3.1. Physical Properties

Figure 2 shows the coat weight as a function of coating thickness for Slurry A, Slurry B, and calendered samples from Slurry B, and the density of solid graphite as a reference.

Figure 3 shows a transmission light test for strips from each roll. A and C show the coating in ambient light conditions for strips in lengthwise and widthwise directions, respectively. B and C show the coating in transmitted light conditions for strips in lengthwise and widthwise directions, respectively. Samples 03A, 03B, 06, and 07 are on the Advantage Kraft Plus paper and 08, 10, 11 and 12 are on the Advantage Boost HP paper. In ambient light conditions, all lengthwise (A) samples, except roll 10, show a fairly even coating. Roll 10 has a spotted surface caused by insufficient drying and thus wet winding. In the widthwise samples, in ambient light conditions (C), a paper distortion feature can be seen. This feature is referred to as cockling and appears as ripples or wrinkles in areas with relatively high coat weight. Roll 08 and roll 11 have the least of this paper distortion. In transmitted light conditions, the quality of the coating can be seen. Roll 03B and UM in the lengthwise direction in transmitted light (B) conditions show clear strip patterns, while roll 08 and roll 11, with the thickest coatings, show a uniform coverage with almost no transmitted light passing through the coating.

Figure 4A shows a tape test of the adhesion of the coating nanographite flakes. The top part of the image shows the pulled-off tape with a white background, and the lower part of the image shows the resulting mark on the strips from the tape for each roll. The tape tests show that only the top nanographite layer peels off with the tape and the adhesion appears to be similar for all samples. Figure 4B shows an SEM image on a cross-section of roll 11 where the red line marks the boundary between the coating (upper part) and the paper substrate (lower part). Roll 11 had the highest coat weight of all rolls, and the cross-section image shows a quite dense and mostly uniform coating. Figure 4C,D show SEM images of the surface of the uncoated Boost HP paper and the surface of the coated paper in roll 11, respectively. The uncoated Boost paper is a machine-finished paper, which is confirmed by the surface image (C) showing a smooth surface with compressed fibers; the image shown is the outside of the paper roll, i.e., the side that later was coated. From the surface image (D) of the coated paper, the coating appears thick with a smooth compressed surface and visual imprints caused by fibers on the paper backside during winding.

### 3.2. Electrical Properties

Figure 5 shows coating resistivity as a function of coating density for all samples made from Slurry A and Slurry B. Calendered samples of Slurry B are also included together with a graphite model representing the theoretical resistivity for solid graphite at different densities, assuming matching homogeneous porosity. From the diagram, it can be seen that the samples follow the same trend as the graphite model, where resistivity decreases with increasing coating density. The samples from Slurry A have relatively large variation, while the samples from Slurry B more clearly follow this trend.

Figure 6 shows coating resistance as a function of distance from the paper edge in both lengthwise and widthwise directions from roll 06 and roll 11. Three strips were measured for each roll in the widthwise direction and one in the lengthwise direction. For both roll 06 and roll 11, each sample strip shows a linear increase in resistance with distance, indicating a uniform coating in both lengthwise and widthwise directions for both rolls. Resistivity was calculated via linear regression, and for roll 06, the resistivity in the lengthwise direction was L1 = 2.42(14) mΩ·m, and in the widthwise direction, it was W1 = 3.88(25) mΩ·m, W2 = 4.52(29) mΩ·m, and W3 = 3.81(17) mΩ·m. For roll 11, the resistivity in the lengthwise direction was L1 = 0.958(38) mΩ·m, and in the widthwise direction, it was W1 = 1.062(37) mΩ·m, W2 = 1.004(35) mΩ·m, and W3 = 0.999(37) mΩ·m. Roll 06 shows anisotropic behavior with a higher resistivity in the widthwise direction compared to the lengthwise direction, while roll 11 shows near-excellent isotropic behavior and also overall lower resistivity.

### 3.3. LIB Anode Application

Roll 03B and roll 11 were further used as anodes for LIB coin cells in a half-cell configuration. Galvanostatic charge–discharge measurements of these cells were made to test the storage capacity of the electrode, whereas cyclic voltammetry measurements were made to study the reaction kinetics of the electrodes. Figure 7A shows a cyclic voltammogram for 100 cycles on a cell from roll 03B, and specific capacity from GCD as a function of cycle number is shown in Figure 7B. Cells B3-1, B3-2, B4-0, and B4-2 are identical, made with anodes from roll 03B, and have an electrode mass of 0.882(62) mg, while cells B5-2 and B6-2 are made from roll 11 with an electrode mass of 0.882(62) mg. From the cyclic voltammograms, it can be seen that both peak heights and area enclosed by the graph increase with the number of cycles. From GCD measurements, it can be seen that the specific capacity of the best-performing cell B3-2 increases linearly up to 100 cycles, reaching 147 mAh/g, and then remains stable until 300 cycles; this corresponds to 40% of the theoretical limit of 372 mAh/g of graphite electrode materials. After 300 cycles, the capacity slightly decreases with further cycling, and at 500 cycles, the capacity has decreased by 8.8%, corresponding to 134 mAh/g. Overall, the capacity for this cell remains fairly stable for extended cycling; the rest of the cells show unstable behavior. Figure 7C shows the corresponding cyclic voltammogram for the reference cell R1; the major anodic peak at 0.18 V is seen to be positioned to the left of the corresponding peak in the B4-0 cell at 0.24 V, a difference of 60 mV; a similar offset towards lower voltages is also seen for the other signatures on the cathodic side of the diagram as well. Figure 7D shows the results of the paper separator’s influence and for the fully charged cell at 1.5 V; the resistivity of 241 Ωm is similar to the 237 Ωm found for an uncharged cell.

## 4. Discussion

### 4.1. Physical Properties

The stability of operations was improved with the Boost paper substrate, and rolls 08 and 11 show no obvious wrinkling or “cockling”; see Appendix A. The physical appearance of the coated surfaces was also different between Slurries A and B when observed by SEM; this can be linked to the two different graphite sources as well as the exfoliation procedures used. It was found that Slurry A has smaller flake size dimensions compared to Slurry B, by about ten times; see Appendix A for further details. Regarding the evenness of coating, striping is seen in Slurry A samples, i.e., roll 03B and UM; visually, it is a line pattern in the traveling direction of the paper during coating, which is an artifact from the LAS coating technology. With better parameter settings and increased coating thickness, these lines disappear and coating becomes more uniform overall, and this is seen for the other rolls of Slurry B; see Figure 3D. In the same figure, one can also see that the coating thickness of roll 06 is not sufficient and the coating looks largely uneven; this is caused by the coating technology in that the valleys of the paper surface are filled first, while the high mountain point, i.e., on top of the outmost positioned fibers in the paper, remains exposed if the coating thickness is too thin. This LAS coating method gives the option of large variations in thickness; an order of magnitude span was achieved during these trials, and when complemented with an extra calendering post-processing step, the density of the coating was increased to 1.12 g/cm^3^ and was very close to solid graphite levels, i.e., 2.25 g/cm^3^, as seen in Figure 2. For comparison, standard LIB 18,650 cells have a coating thickness typically between 30 μm and 90 μm and an electrode density around 1 g/cm^3^ [28]. In roll 11, the highest achieved coating thickness was 47.7(1.4) μm with a coat weight of 17.65(29) g/cm^2^, indicating that the coating technique allows for sufficient coating thicknesses. Considering that coat weight is strongly dependent on the slurry’s solids content, it is likely possible to also achieve equivalent sufficient coat weights with an improved slurry and further optimized coating operations. Besides the slurry itself, the combination of coating parameters together with the substrate properties is crucial to reach a specific electrode thickness or density. The amount of wet material (slurry) that is applied to the paper substrate can be varied by adjusting the machine speed, transfer roll speed, and metering nip, but the paper properties (e.g., absorption rate and wet expansion) affect the web tension in the coating process, which affects the calendering effect (compression) on the electrode from the rolls before winding. The drying conditions are not assumed to have the same effect on electrode density but still need to be adjusted according to the wet coat weight to ensure that sufficient drying occurs without overheating the paper or slurry. Electrode density is important for electrodes used in battery applications, where a compact electrode for small ions is preferred. Hence, this coating method allows for tailoring the electrode density for different types of application. The LAS coating technology used is directly scalable to industrial levels. LAS can be scaled up to paper widths exceeding 6 m for large-scale production, and the maximum machine speed is above 2000 m/min. In a possible maximum scale-up for roll 11, the machine speed could be increased by a factor of 7.5 (limited by the speed of the transfer roll), and with a 6 m coating width, the production rate would reach 900 m^2^/min in a single production line. In the traditional industrial LIB cell production process, the coat speed is in the range of 35 m/min to 80 m/min with a coating width of up to 1.5 m [29]. To the best of our knowledge, no study has been published on the large pilot-scale coating of paper for electrode applications. However, for comparison of the achieved scale-up performances of our work and to visualize the advantages in the speed of area production by utilizing coating methods on top of existing substrates, it can be clearly seen that the small pilot trial that demonstrated a paper conversion approach and was conducted at a web width of 20 cm and a speed of 1 m/min is slower [19]. For a more proper comparison with similar materials, the spray-coating approach in a paper fabrication process at a large pilot scale achieves results in the same range but somewhat slower with 15 m/min, a similar web width, and a lower electrode/spray weight of 11.6 g m^2^, although how a further scale-up of this spray method from pilot to industry with a preserved consistent quality over web widths is supposed to be carried out is uncertain [22].

### 4.2. Electrical Properties

Anisotropic behavior of resistivity is favorable in some applications, for example, in thermal applications of conduction of heat since heat transfer is linked to electrical transport phenomena; however, in this study, the focus was on achieving isotropic behavior, which was also demonstrated for the thicker coatings like roll 11, as shown in Figure 6. The threshold for isotropic behavior is roll 07, i.e., a coat weight of 11.42 g/m^2^; all samples with a lower coat weight than that showed anisotropic behavior. This anisotropic behavior can be explained by two things: First, the LAS coating technique can influence it, as seen in the Slurry A trials, and if not adjusted properly, when coatings are thin, it forms strip patterns across the width of the roll, leading to very high anisotropic behavior in resistivity. Secondly, the anisotropy of the paper favors the fibers aligning in the lengthwise direction during paper manufacturing, leading to a smoother surface and hence, for a thin coating, keeping an electrical connection between the nanographite flakes is easier compared to going across fibers in the widthwise direction. For more details on this, see Appendix A.

It is worth mentioning that the method used for measuring the resistivity also provides an excellent way to characterize the overall coating quality; thus, when making resistance vs. length measurements, one can clearly see if there is a coating defect and, for example, if the coat weight changes across the width of the paper, which can happen with improper setting of the LAS system; this is seen in roll 06 in the left graph of Figure 6 for the W1 strip, showing a nonlinear trend with a decreased slope in the data curve towards the edge of paper, indicating that coat weight increased by about 1/3.

In Figure 5, two results can be observed. Firstly, the best performance was found in the calendered sheets, giving values close (50%) to the theoretical limit of porous graphite structures with similar densities to the coating. This indicates that the connections between the nanographite flakes are overall good and are not hampered by the added dispersive agent during slurry mixing. Secondly, Slurry B outperforms Slurry A in that the former is in all coating trials closer to the theoretical limit of graphite. This could perhaps be explained by the difference in nanographite flake size and the fact that Slurry A has smaller sizes compared to Slurry B, and hence, hypothetically, if the same coating density is achieved in two coatings but different slurries are used, then Slurry A will have more interconnected electrical resistance points between the flakes since the number of flakes becomes larger to compensates for their smaller sizes; see Appendix A for detailed images of flake morphology. Overall, a correlation study like Figure 5 is a good method to characterize the material. Since the resistivity dependency of density says something about the internal structure of the material and since we have two flake size distributions in our experiments, we can see that this is also observed as a feature in that same figure; a future investigation could be to correlate this resistivity change with the different dispersive agents used.

### 4.3. LIB Anode Application

The cyclic voltammograms (Figure 7A) show an increase in the peak height of the CV curve during the cycling. A higher peak indicates more lithium ions being inserted into the anode. Higher peaks suggest that a larger portion of the anode material actively participates in the electrochemical reaction overall. However, it is observed that for the paper electrode, this process of activating the electrode is rather slow compared to more standard cells, and we do not see a stabilization in its performance until 100 cycles.

The paper electrode has all its peaks offset to higher voltage values compared to the reference cell R1’s CV data (Figure 7C). For the highest peak on the anodic side, this offset is about 60 mV; this is comparable to the influence of the paper separator’s voltage contribution due to added series resistance. Measured (Figure 7D) with a resistivity of 241 Ωm, a single paper layer as used for cell B4-0 will correspond to an added series resistance of 116 Ω, and at the peak current of 165 μA, this will add a voltage offset of 19.1 mV. If compared with R1, one has to also take into account the voltage contribution of the Celgard membrane as well, but this has a much lower value than the paper’s contribution, an estimate since the membrane is 25 μm in thickness, which is four times thinner than the paper, and with the approximation and all else being equal, this contribution is about 4 mV and the offset between the two cells caused by separator influences will be 15 mV. Hence, the observed offset could be explained as being at least partly caused by the paper’s higher electrical resistance.

The long-term stability of battery capacity over extended cycling highlights the potential of using paper as a separator. Furthermore, this also strengthens the potential of a new design concept wherein paper can act as both a separator and a substrate for coating the anode material. A unique advantage of this approach is that the electrode is electrochemically accessible from both of its sides since the graphite in this case is not coated onto a metallic foil, which is the more traditional approach.

#### Other Applications of the Coated Paper Electrode

The early coating experiments prior to the content of this study have shown that triboelectric nanogenerators utilizing coated electrode paper from another batch of Slurry B on Kraft Plus paper can generate a peak power density of 14 kW/m², driven by the electrostatic discharge effect on the material’s surface [25]. The same study also showed that the coated paper could be used for wall- or floor-mounted smart sensors to detect movement as well as smart sheets that monitor body movements and physiological activities during sleep. Furthermore, lab-scale coating experiments prior to this study have shown that other slurry batches made with the same equipment coated on paper can also be used as electrodes in aqueous, metal-free, and low-cost supercapacitors generating a specific capacitance of 48 F/g [30].

## 5. Conclusions

This study revealed the following two conclusions. First, it was demonstrated that a functional electrically conducting material coated onto a paper substrate could be produced at a pilot scale (TRL6) with roll-to-roll methods and that this material shows promising performance for further electrode applications, especially since it is environmentally friendly, resource-efficient, and low-cost. The best values achieved in this study were a coat weight of 17.65(29) g/m^2^ (roll 11), a max coating thickness of 56.1(2.5) μm (roll 12), the highest density of 1.117(97) g/cm^3^ (roll 08), and the highest electrical conductivity of 0.1293(17) mΩ·m (roll 08), making roll 08 the best-performing coating on average. Secondly, the two characterization methods used to measure resistance as a function of a dimensional distance and study resistivity as a function of density are both good tools for further materials science investigations when characterizing fundamental properties of new materials at the nanoscale.

## Figures and Tables

**Figure 1 nanomaterials-15-00113-f001:**
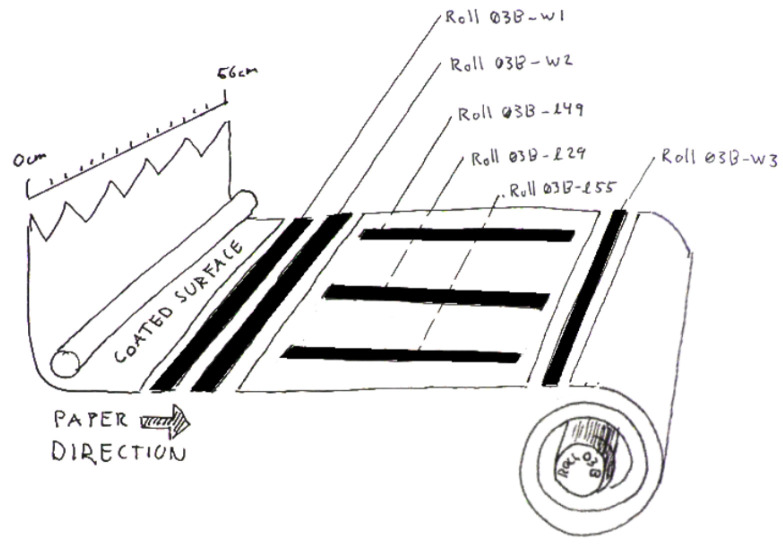
Schematic sketch of how sample strips were cut from rolls of coated paper, with examples of how the samples were named.

**Figure 2 nanomaterials-15-00113-f002:**
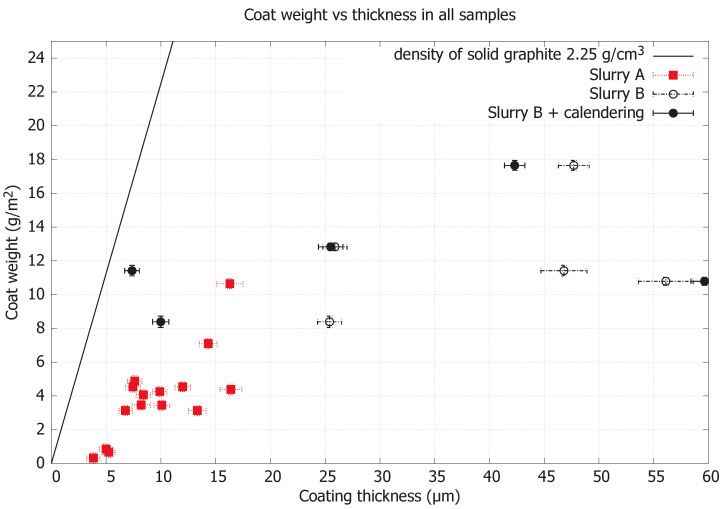
Coat weight as a function of coating thickness for Slurry A, Slurry B and calendered samples from Slurry B. The solid line represents the boundary when the density of coating becomes the same as that of solid graphite.

**Figure 3 nanomaterials-15-00113-f003:**
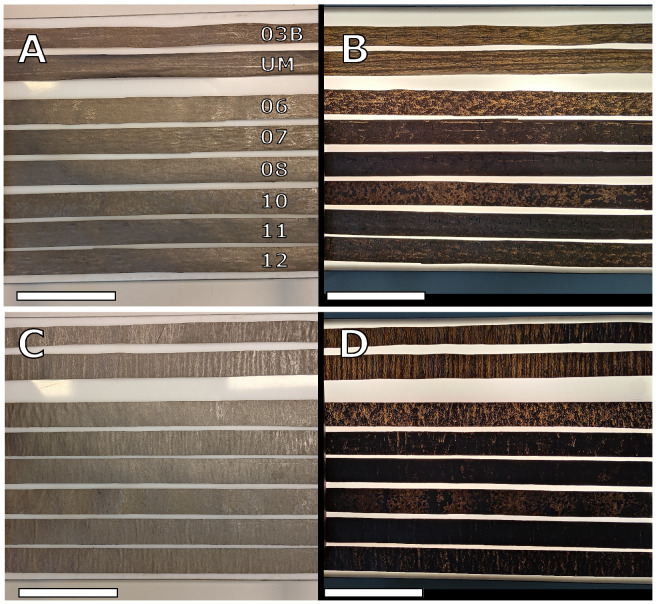
Transmission light test for strips from each roll. (**A**) shows the coating in ambient light conditions for strips in the lengthwise direction and in (**B**), the same strips are shown under transmitted light conditions. (**C**) shows the coating in ambient light conditions for strips in the widthwise direction, and in (**D**), the same strips are shown under transmitted light conditions. Scale bars are 10 cm.

**Figure 4 nanomaterials-15-00113-f004:**
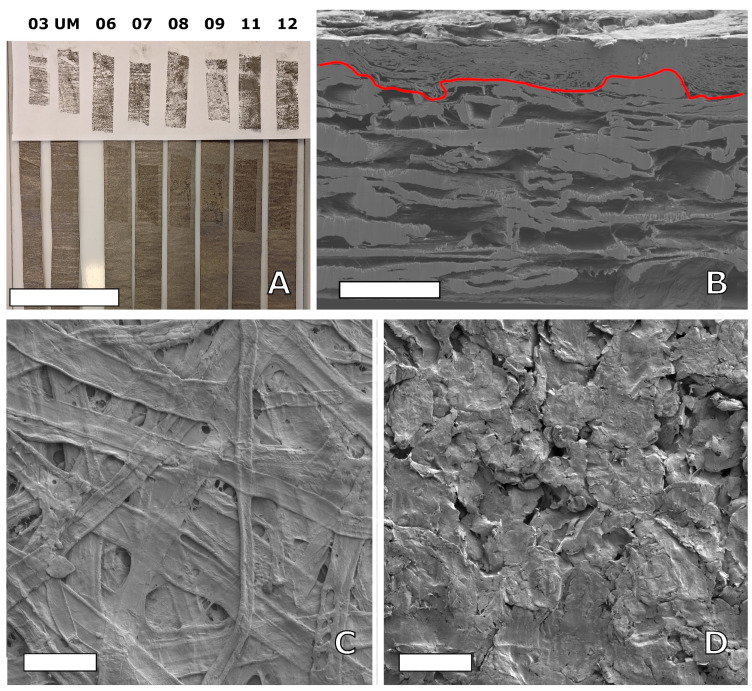
(**A**) Adhesive test by tape pulling method. The top part shows that the electrode remains stuck to the tape, and the bottom part shows the location where the tape was placed and pulled; the scale bar is 10 cm. (**B**) SEM (SE) image of a cross-section from roll 11; the red line marks the boundary between the coating at the top and the paper substrate at the bottom; the scale bar is 50 μm. (**C**) SEM (SE) top-view image of uncoated Boost HP paper, and (**D**) top-view image of the coated roll 11; the scale bars are 100 μm in (**C**,**D**).

**Figure 5 nanomaterials-15-00113-f005:**
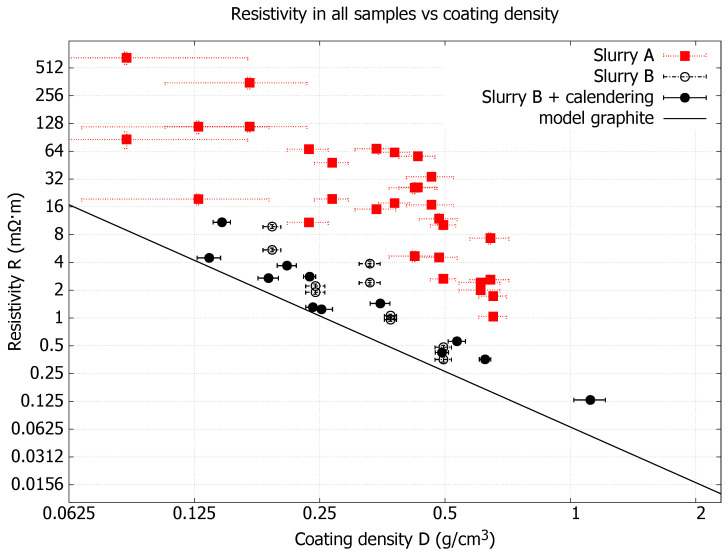
Coating resistivity as a function of coating density for all samples made from Slurry A and Slurry B. Calendering refers to sheets taken from coated rolls that were calendered without heat. The graphite model represents the theoretical resistivity for solid graphite at different porosities (densities). Error bars are standard deviation errors.

**Figure 6 nanomaterials-15-00113-f006:**
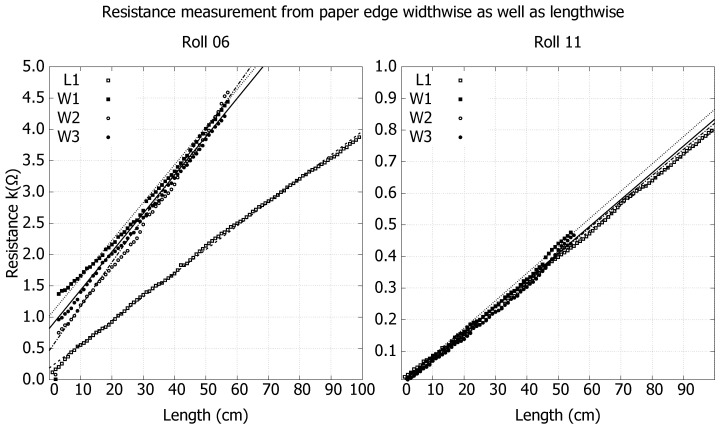
Resistance as a function of length from the edge on three strips cut in the widthwise direction and one strip in the lengthwise direction from roll 06 and roll 11. For roll 06: L1 = 2.42(14) mΩ·m, W1 = 3.88(25) mΩ·m, W2 = 4.52(29) mΩ·m, and W3 = 3.81(17) mΩ·m. For roll 11: L1 = 0.958(38) mΩ·m, W1 = 1.062(37) mΩ·m, W2 = 1.004(35) mΩ·m, and W3 = 0.999(37) mΩ·m.

**Figure 7 nanomaterials-15-00113-f007:**
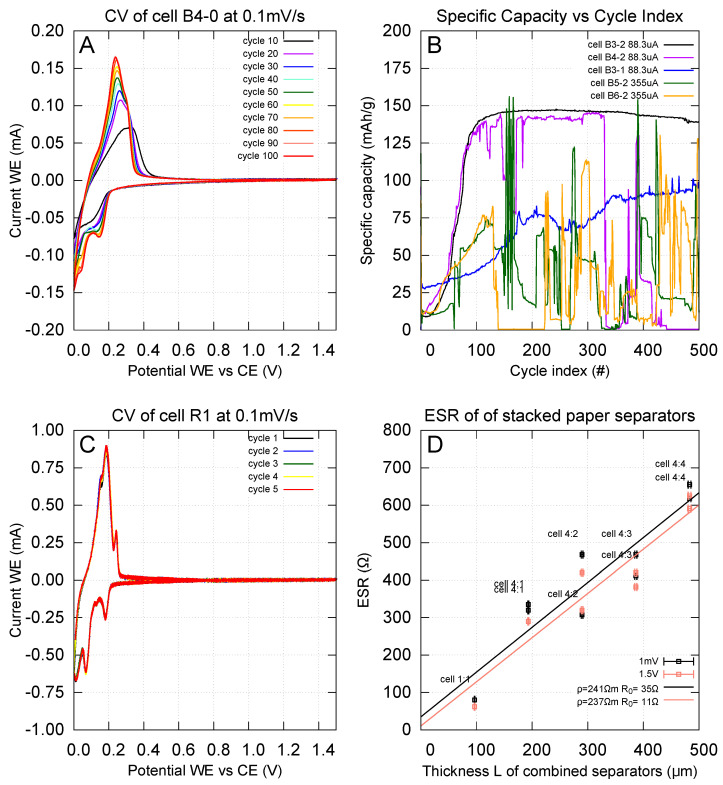
Cyclic voltammetry (**A**) on a half-cell configuration made from roll 03B and run for 100 cycles (35-day run time) at a rate of 0.1 mV/s. Galvanostatic charge–discharge cycling (**B**) for five half-cells; the total run time for cell B3-2 is 50 days for 500 cycles. Cells B3-1, B3-2, B4-0, and B4-2 are all identical and made with electrodes from roll 03B, corresponding to an electrode mass of 0.882(62) mg, while cells B5-2 and B6-2 are made from roll 11, mass 0.882(62) mg. Cyclic voltammetry (**C**) on reference cell R1, a graphite-coated copper foil instead of paper. Comparison (**D**) of electric series resistance (ESR) conducted by inserting additional sheets of paper as extra separator material, each 96.60(50) μm, showing that the resistivity of the Advantage Kraft Plus paper is 241 Ωm for a cell in a fully charged state.

**Table 1 nanomaterials-15-00113-t001:** Summary of the paper electrode rolls and main coating parameters. Values in brackets are the standard deviation error of the last two significant digits of the measurement values. Coating density is a calculated value based on measured thickness and weight.

Roll Number	Coating Slurry	Substrate Paper	Machine Speed [m/min]	TR Speed [m/min]	Metering Nip [mm]	Coat Weight [g/m^2^]	Coating Thickness [μm]	Coating Density [g/cm^3^]
UM	A	Plus	25	−700	35	4.07(28)	8.41(66)	0.48
03B	A	Plus	25	−900	38	4.39(31)	16.4(1.0)	0.27
06	B	Plus	25	−400	38	8.39(33)	25.4(1.1)	0.33
07	B	Plus	25	−800	38	11.42(31)	46.8(2.1)	0.24
08	B	Boost	25	−800	38	12.83(22)	25.9(1.1)	0.50
10	B	Boost	15	−500	28	13.19(75)	38.3(2.8)	0.34
11	B	Boost	20	−400	35	17.65(29)	47.7(1.4)	0.37
12	B	Boost	25	−400	38	10.79(24)	56.1(2.5)	0.19

## Data Availability

Data are contained within the article or Appendix A.

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
