# Peer review of "Large-Scale Compatible Roll-to-Roll Coating of Paper Electrodes and Their Compatibility as Lithium-Ion Battery Anodes"

_nanomaterials, 2025, doi:10.3390/nano15020113_

Round 1

Reviewer 1 Report

Comments and Suggestions for Authors

The manuscript by N. Blomquist et al. reported the use of roll-to-roll coating technology and paper-based electrodes for sustainable LIB design. The manuscript addresses a timely and important topic on sustainable energy storage materials. The conclusions are well-supported by their experiment design and results, such as the coat weight, thickness, and electrical resistivity measurements. However, the discussion could benefit from a deeper comparison with existing technologies and clearer implications for scalability and industrial adoption. Here are some detailed comments to the authors:

1. The methods for preparing the two slurries (A and B) are clear, but additional details on the differences in flake size and how these might influence coating uniformity and resistivity would improve clarity.

2. Figure 3 could include more specific labels (e.g. slurries, thickness, coating times) to make it accessible to readers unfamiliar with the technique.

3. The anisotropy observed in the resistivity measurements is interesting but the authors should use additional analysis or literature comparisons to explain the reasons and the strategy to avoid such phenomenon.

4. The discussion of the reasons for slow activation process and large capacity fluctuation in some of the electrodes would help to explore possible solutions for better coating technique in future work.

5. Labeling of axes and color coding of the curves in figures 6 and 7 should be improved for better readability.

6. The authors should clarify the role of specific parameters (e.g., calendering, drying conditions) in improving coating density and electrical conductivity, as these insights are crucial for industrial adaptation.

7. I would recommend the authors to Include a more detailed discussion on the industrial scalability of the roll-to-roll process, focusing on challenges like cost, waste management, and reproducibility at higher speeds, emphasizing the practical implications and broader impact of the work on the battery industry.

Author Response

Comment 1: The methods for preparing the two slurries (A and B) are clear, but additional details on the differences in flake size and how these might influence coating uniformity and resistivity would improve clarity.
Response 1: We agree with this, but have not done any any flake size characterization on these two slurries, as it is a very time-consuming work to do to reach a sufficiently reliable result. This has been done previously for the process used for slurry B (cited in the Electrode material preparation section), but then the slurry was processed at a lower graphite concentration so the repeatability cannot be guaranteed. However, in the SEM images shown in Supplementary Figure 6 (S6) one can clearly see differences in particle size, where the flake size is estimated to be about 10 times larger in slurry B compared to Slurry A. We have clarified this by adding the following text to the figure text of S6: “Except for the thinner coating on roll 03B compared to roll 11 the other observations seen are differences in flake size where approximately ten times smaller flakes are present in slurry A, as seen in the detailed image (A) of roll 03B, compared to (C) made from Slurry B.”
We have also added the size difference approximation to the discussion section of the manuscript under Physical properties where the physical appearance of the coated surfaces are discussed, with the following text: “The physical appearance of the coated surfaces was also different between the two slurries A and B when observed by SEM, this can be linked to the two different graphite sources as well as the exfoliation procedures used, it was found that slurry A has smaller flake size dimensions compared to slurry B, about ten times, see supplementary figure~S6 for further details.”

Comment 2: Figure 3 could include more specific labels (e.g. slurries, thickness, coating times) to make it accessible to readers unfamiliar with the technique.
Response 2: We have actually tested this but decided to remove does details from figure 3 to increase the clarity of the main differences of each slurry A, B. Full details is still accessible in the full detailed table in the supplementary.

Comment 3: The anisotropy observed in the resistivity measurements is interesting but the authors should use additional analysis or literature comparisons to explain the reasons and the strategy to avoid such phenomenon.
Response 3: We believe this is not necessary since it is already described in the discussion section. This is due to two identified phenomena’s, antistrophic paper and the stripe patterning effect of the LAS coating system. Hence, the better formula and parameter settings that we demonstrated also showed a good solution to the problem.

Comment 4: The discussion of the reasons for slow activation process and large capacity fluctuation in some of the electrodes would help to explore possible solutions for better coating technique in future work.
Response 4: We agree on this and this will be investigated in future work.

Comment 5: Labeling of axes and color coding of the curves in figures 6 and 7 should be improved for better readability.
Response 5: In figure 6 the number of data points is large and the main idea with these graphs is not the data details but the overall differences in anisotropy, which we believe is best shown in the way its presented now were the data set are distinguished by different point types.

In figure 7, we already have labels and color coding, as proposed, so it’s not clear for us what’s seems to be missing here.

Comment 6: The authors should clarify the role of specific parameters (e.g., calendering, drying conditions) in improving coating density and electrical conductivity, as these insights are crucial for industrial adaptation.
Response 6: Thank you for pointing this out, we agree and the manuscript is missing a description for this on the electrode density. The connection between electrode density and electrical conductivity are already described but we have added the following text to the discussion section, under Physical properties: “Besides the slurry itself, the combination of coating parameters together with the substrate properties are crucial to reach a specific electrode thickness or density. The amount of wet material (slurry) that is applied to the paper substrate can be varied by adjusting the machine speed, transfer roll speed and metering nip, but the paper properties (e.g. absorption rate and wet expansion) affect the web tension in the coating process, which affects the calendering effect (compression) on the electrode from the rolls before winding. The drying conditions are not assumed to have the same effect on electrode density but still needs to be adjusted according to the wet coat weight to ensure that sufficient drying occurs without overheating the paper or slurry.”

Comment 7: I would recommend the authors to Include a more detailed discussion on the industrial scalability of the roll-to-roll process, focusing on challenges like cost, waste management, and reproducibility at higher speeds, emphasizing the practical implications and broader impact of the work on the battery industry.
Response 7: We agree on this and have added the following text to the discussion section, under Physical properties:
“The LAS coating technology used is directly scalable to industrial level. LAS can be scaled up to paper widths exceeding 6 m for large scale production and maximum machine speed is above 2000 m/min. In a possible maximum scale-up for roll 11 the machine speed could be increased by a factor of 7.5 (limited by the speed of the transfer roll) and with 6 m coating width the production rate would reach 900 m2/min in a single production line. In a traditional industrial LIB cell production process the coat speed is in the range of 35 m/min to 80 m/min with a coating width up to 1.5 m {heimes2018lithium}.”

Reviewer 2 Report

Comments and Suggestions for Authors

This manuscript reports on a large-scale roll-to roll coating method for fabrication of graphite coated paper electrodes for lithium-ion batteries. Overall, this work is well conducted with convincible results. I suggest the publication of this paper after solving the following issues.

(1) The motivation of this work is to fabricate electrode using biomaterial such as paper and cellulose to improve the recyclability of lithium-ion batteries. However, when assembling the lithium-ion cells for testing, copper foil is still used as the anode current collector. Herein, why not directly coat the graphite on copper foil and use the paper as separator?

(2) Figure 7 B, the authors should comment on the huge differences in both specific capacities and cycling stability among the different electrodes being tested.

(3) What is the specific capacity of the graphite being coated on copper foil as shown in Figure 7C. This capacity should be reported and compared with those from the paper electrodes.

(4) The supplementary information should be organized into a single word/PDF file.

Author Response

Comment 1: The motivation of this work is to fabricate electrode using biomaterial such as paper and cellulose to improve the recyclability of lithium-ion batteries. However, when assembling the lithium-ion cells for testing, copper foil is still used as the anode current collector. Herein, why not directly coat the graphite on copper foil and use the paper as separator?
Response 1: Thank you for pointing this out, as it may not have been clearly described in the first version of the manuscript. To limit the comparison to only studying the difference between an electrode casted directly on the copper foil (as the reference cells) and the paper-coated electrodes, copper foil was added as a “loose” current collector in the coin cells. Without copper foil in a coin cell, the stainless-steel casing of the coin cell becomes the current collector, which can cause differences that are not related to the electrode or separator itself. How the choice of current collector and contacting affects the battery application is something that needs further research in future studies.

To clarify this in the manuscript, we have added the following in the Materials and Methods section, under Lithium-ion anode application:
“Another 16~mm disc of copper foil was added as a current collector towards the nanographite (anode) side, to allow comparison with traditional reference cells without influence from different electrode-contact interface materials.”

When asked why not directly coat copper foil, there are several answers. First of all, coating on paper substrates allows a significantly higher coating rate (machine speed), as the substrate itself absorbs the solvent in the slurry and drying may occur from both the bottom and top. This is already done today with other types of slurry in the selective paper industry. The type of coater used in this study is capable of coating paper substrates at over 2000 m/min and up to 6 m width. Lastly, a long-term aim of our development of technology is to overall realize lower resource usage, higher cost efficiency, and better recyclability and this by removing of the traditional metallic current collector with a post processing of the graphite coatings backside, this could for example be realized via different types of laser processing methods. 

Comment 2: Figure 7 B, the authors should comment on the huge differences in both specific capacities and cycling stability among the different electrodes being tested.
Response 2: The huge differences in specific capacitance we already commented on in the discussion section. The abrupt changes we see in performance difference of the cells shown in 7B, we identify as a mechanical failure of the electrical connections within the electrode material. Besides this we haven't made further studies on these artifacts and it will be subjected to future research.

Comment 3: What is the specific capacity of the graphite being coated on copper foil as shown in Figure 7C. This capacity should be reported and compared with those from the paper electrodes.
Response 3: We understand that it could be an interesting point to report on specific capacity for the reference cell. However, the purpose of the reference cell in this study was to be able to compare the peak offset to different voltage values for the paper electrodes compared to the reference cell CV data, as we also discuss in the manuscript.

It is also less fruitful in doing as you propose a direct comparison between two single cells capacity performances and this since one must consider the measurement error influence caused by the low coating mass vs mass of the copper foil ratio. An estimate in this particular scenario will lead to a 79% error of specific capacitance if such a comparison is being done and hence, not particularly scientific. However, we plan in future work to investigate battery properties more separately and decided this manuscript to focus on the coating technology aspect mainly.

Comment 4: The supplementary information should be organized into a single word/PDF file.
Response 4: We agree on this and the supplementary file will be converted into a single pdf in a later stage. For the first submission both manuscript and supplementary was attached as Latex files made according the journal template.

Reviewer 3 Report

Comments and Suggestions for Authors

This is a very interesting work for me. The authors replaced the synthesized polymeric separator (PP) with Kraft paper. The main composition of Kraft paper is CMC, which can be decomposed easily in nature. I suggest the publication of this work. I have some suggestions and questions hereï¼›

1.  In the Abstract, it is better to add the cycling condition such as rate and cutoff voltage for the cycling capacity of the battery.

2. It is better to test the chemical stability and composition change in the slurry solution and electrolyte, respectively. I think the dissolution of impurity in the Kraft paper may impact performance. For example, in Chemical Engineering Journal 437 (2022) 135283, the 0.05 wt%  additive in ester-based electrolyte significantly impacts performance. The author can investigate the impact of the impurity of the Kraft paper in future work.

3. What is the thickness of the separator? Please mention that in the experiment of the manuscript.

4. It would be better if the authors draw a schematic illustration of the battery structure and the preparation processes to make the manuscript read easily.

5. Since the contact between graphite coating and the current collector is not good for this configuration, the electrical conductivity is bad compared to the traditional process. The good contact may need high pressure to achieve that, but high pressure may lead to easy short circuit of the battery.

6. Considering the thickness and porosity, I think PP separator still is the best option for commercial batteries. But this work is a good start.

Author Response

Comment 1:  In the Abstract, it is better to add the cycling condition such as rate and cutoff voltage for the cycling capacity of the battery.
Response 1: Thank you for commenting on this but unfortunately, we advise against doing so. By adding a lot of technical details in the abstract it tends to counteract the purpose of a short and clear abstract. Details is anyway presented in the main part of the manuscript.

Comment 2: It is better to test the chemical stability and composition change in the slurry solution and electrolyte, respectively. I think the dissolution of impurity in the Kraft paper may impact performance. For example, in Chemical Engineering Journal 437 (2022) 135283, the 0.05 wt%  additive in ester-based electrolyte significantly impacts performance. The author can investigate the impact of the impurity of the Kraft paper in future work.
Response 2: Thank you for pointing this out, this is a good comment. We are already doing future work in this direction.

Comment 3: What is the thickness of the separator? Please mention that in the experiment of the manuscript.
Response 3: Thanks for pointing this out, we have added the paper thickness in section 2.2 Electrode coating as well as the following figure text to figure 7D: " Comparison (D) of electric series resistance ESR of inserting additional sheets of paper as extra separator, each 96.60(50) um, showing that the resistivity of the Advantage Kraft Plus paper is 241 Ωm for a cell in full charge state"

Comment 4: It would be better if the authors draw a schematic illustration of the battery structure and the preparation processes to make the manuscript read easily.
Response 4: We agree, we have added a schematic illustration of the coin cell battery structure to supplementary figure 2 (S2 I). Figure 1 and 2 in supplementary (S1 och S2) now gives a good visual overview of all parts of the material and sample preparation processes.

Comment 5: Since the contact between graphite coating and the current collector is not good for this configuration, the electrical conductivity is bad compared to the traditional process. The good contact may need high pressure to achieve that, but high pressure may lead to easy short circuit of the battery.
Response 5: During our calendaring studies, which implies compressing both the electrode and the paper, we have not seen such problems with separator penetration. We haven't done follow up complete studies on this aspects since we decided in this manuscript to report on the main findings of the coating procedures.

Comment 6: Considering the thickness and porosity, I think PP separator still is the best option for commercial batteries. But this work is a good start.
Response 6: We agree that this is overall correct from a performance perspective, from a sustainability perspective PP materials is less favorable. Also as seen in the work done by Raphael Zahn et al. (10.1021/acsami.6b12085) that theoretically one can expect a MacMullin number of  6.8 for a separator (in which a low number is better) but regular membranes tend to come in at much higher values. For example a PE separator has an number of 20 (Raphael Zahn et al.)  while a paper based separator achieves values in the range of 3-6 (Lorenzo Zolin et al. 10.1039/C5TA03716F). This concludes that a further development of paper based separators mainly to achieve thinner versions could be most favorable even performance wise as well.

We have also added this to the introduction section with the text: “LIB PP or PE separators are usually less than 25 μm thick sheets with 40% porosity and sub-micrometer pore sizes. Using paper as separator has been shown to be advantageous even in terms of performance, as these have proven MacMullin numbers (ratio of ion conductivity between a wetted separator and the ion conductivity of the free electrolyte) of 3-6 {zolin2015aqueous} which can be compared with typically 20 for a PE separator {zahn2016improving}.”

Reviewer 4 Report

Comments and Suggestions for Authors

In the manuscript, the authors present an innovative recyclable design for LIBs by utilizing paper-based electrodes and separators. The relationship between resistance and density is investigated, and roll-to-roll fabrication has been demonstrated at a pilot scale. While the study shows promising electrical properties and battery performance, further optimization of the slurry composition and coating parameters is expected to improve conductivity and capacity. Detailed comments are provided below.

1. More studies on paper-based electrodes by coating or mixing techniques should be cited and discussed, as the focus of this work is on electrode design rather than battery recycling. Furthermore, a proper comparison of the physical properties and battery performance is recommended.

2. It is noticed that copper foil is still used as the current collector, so it is not a fully paper-based design. Do the electrodes provide significant advantages in recycling, or are they ultimately disposed of? A brief life cycle assessment and cost analysis is suggested.

3. The coating weight is relatively lower than that of commercial electrodes (typically over 50 g m-2). What are the potential limitations in increasing the loading and density? The authors should clarify the parameters in Table 1 more thoroughly to better explain the differences in uniformity observed in Figure 3.

4. The resistivity and capacity are not satisfying to the market. The paper layer appears quite thick in Figure 4B, which may contribute to reduced capacity and energy efficiency. Additionally, the curves in Figure 7B are noisy. What is the possible reason for it? 

Author Response

Comment 1: More studies on paper-based electrodes by coating or mixing techniques should be cited and discussed, as the focus of this work is on electrode design rather than battery recycling. Furthermore, a proper comparison of the physical properties and battery performance is recommended.
Response 1: We agree on this and have cited three more paper-related studies in the introduction with the following texts:

“LIB PP or PE separators are usually less than 25 um thick sheets with 40 % porosity and sub-micrometer pore sizes. Using paper as separator has been shown to be advantageous even in terms of performance, as these have proven MacMullin numbers (ratio of ion conductivity between a wetted separator and the ion conductivity of the free electrolyte) of 3-6  {zolin2015aqueous} which can be compared with typically 20 for a PE separator {zahn2016improving}, were a lower ratio corresponds to better ion conductivity.”

“To make a paper-based electrode the paper itself can be converted into an electrode and this has been already demonstrated at pilot scale operations in which a paper where fabricated containing active carbon and PEEDOT:PSS ingredients {brooke2022large-scale, sandberg2016photoconductive}.”

“Another reported method is spray deposition of graphite and microfibrillated cellulose onto bleached softwood pulp prior to pressing and drying in a pilot paper machine. This method show an LIB anode capacity of 95 mAh/g at 1C with an electrode thickness of 27.5 um, specific weight of 11.6 g/m2 and an electrical resistivity of about 500 ohm/sq (about 14 ohm m) {beneventi2014pilot}.”

We also added the following to the discussion section, under Physical properties:
“To the best of our knowledge, no study has been published based on large pilot scale coating of paper for electrode application. However, for comparison of the achieved scale up performances of our work and to visualize the advantages in speed of area production by utilizing coating methods ontop of existing substrates it can be clearly seen that the small-pilot trial that demonstrated a paper-conversion approach and were done at web widths of 20 cm and speed of 1 m/min is slower {sandberg2016photoconductive}. For a more proper comparison with similar materials, the spray-coating approach in a paper fabrication process at large-pilot scale achieves results in the same range but somewhat slower with 15 m/min, similar web width and less electrode/spray weight of 11.6g m2 {beneventi2014pilot}.”

Comment 2: It is noticed that copper foil is still used as the current collector, so it is not a fully paper-based design. Do the electrodes provide significant advantages in recycling, or are they ultimately disposed of? A brief life cycle assessment and cost analysis is suggested.
Response 2: This is correct, the copper foil is added as a “loose” current collector in the coin cells, to allow comparison with traditional reference cells without possible influence from different electrode-contact interface materials. We have now clarified this in the manuscript by adding the following in the Materials and Methods section, under Lithium-ion anode application:
“Another 16~mm disc of copper foil was added as a current collector towards the nanographite (anode) side, to allow comparison with traditional reference cells without influence from different electrode-contact interface materials.”

Not coating directly on the current collector has advantages in the recycling process, as the current collector is "loose" and can be easily removed. Paper substrates and binders (cellulose) can be recycled through existing large-scale processes in the paper industry, to either become new paper, binders or other cellulose-based materials. Recycling of nanographite is not yet fully explored. It can be converted back to graphite by heat treatment but since it’s a low cost material, this process is not yet cost efficient.

A "loose" current collector also provides great freedom to design the contacting as needed, so this can be seen as a intermediate step for a fully metal-free electrode

We believe that a life cycle assessment and cost analysis is not necessary at the moment, according to e.g. Popien JL et al. (https://doi.org/10.1007/s11367-023-02134-4), these are often product-specific and based on existing value chains, making this type of analysis rather uncertain for new concepts like this.

Comment 3: The coating weight is relatively lower than that of commercial electrodes (typically over 50 g m-2). What are the potential limitations in increasing the loading and density? The authors should clarify the parameters in Table 1 more thoroughly to better explain the differences in uniformity observed in Figure 3.
Response 3: This is correct, in standard 18650 cells the anode graphite coating thickness is typical in the range of 30um (LFP) - 90um (NCA) depending on the power/Energy demand  and the electrode density is about 1 g/cm3 giving a a coat weight range about 30-90 g/m2 (Katharina Bischof et al. DOI: 10.1016/j.powera.2024.100148). In our best achieved result we coated 47.7um, and a coat weight of 17.65 g/m2. Although the latter is lower compared to commercial 18650 it still demonstrates that the technology developed is in the same league. In our developed coating method, it is the thickness rather than mass that limits the upper ranges possible when slurry is applied to the paper. The final coating density is governed by slurry properties (e.g solids content) and an improvement on the slurry is seen as way forward to increase weight of coating.

We have added a comment on this in the discussion section, under Physical properties along with a discussion about the coating parameters. The following text is added:
“For comparison, standard LIB 18650 cells have a coating thickness typically between 30 um  and 90 um and an electrode density around 1 g/cm3 {bischof2024evaluation}. In Roll 11 the highest achieved coating thickness was 47.7(1.4) um with a coat weight of 17.65(29) g/cm2 indicating that the coating technique allows for sufficient coating thicknesses. Considering that coat weight is strongly dependent on the slurry's solids content, it is likely possible to also achieve equivalent sufficient coat weights with an improved slurry and further optimized coating operation. Besides the slurry itself, the combination of coating parameters together with the substrate properties are crucial to reach a specific electrode thickness or density. The amount of wet material (slurry) that is applied to the paper substrate can be varied by adjusting the machine speed, transfer roll speed and metering nip, but the paper properties (e.g. absorption rate and wet expansion) affect the web tension in the coating process, which affects the calendering effect (compression) on the electrode from the rolls before winding. The drying conditions are not assumed to have the same effect on electrode density but still needs to be adjusted according to the wet coat weight to ensure that sufficient drying occurs without overheating the paper or slurry.“

Comment 4: The resistivity and capacity are not satisfying to the market. The paper layer appears quite thick in Figure 4B, which may contribute to reduced capacity and energy efficiency. Additionally, the curves in Figure 7B are noisy. What is the possible reason for it? 
Response 4: We agree on this. Here we used standard papers in a new application to show proof of concept. Improvement on paper properties is needed before reaching commercially satisfaction in a high power application, however in low power application such as low-cost grid storage solutions, resource efficiency and long life is dominating and we see this developed technology to target that area at first hand and the others secondly.

The noise in 7B is not really noise in an electrical aspect, we seen this behavior in several cells, it is an abrupt change in capacity which we identify as mechanical instability of the electrode i.e. it loses electrical connection temporarily within its structure. This we commented on in the discussion, but we have not made further investigation on it since we decided to keep the storyline strict in this manuscript focusing on the developed coating method and its applicational usage is seen as secondary. In a future follow-up study we will investigate these battery/energy storage aspects in detail.